

# Treatment gap and barriers to access mental healthcare among women with postpartum depression symptoms in Pakistan

Aqsa Sajjad[1], Shahid Shah[1], Ghulam Abbas[2], Ayesha Aslam[3], Fawad Randhawa[4], Haris Khurram[5,6] and Abdullah Assiri[7]

[1] Department of Pharmacy Practice, Faculty of Pharmaceutical Sciences, Government College University, Faisalabad, Faisalabad, Pakistan

[2] Department of Pharmaceutics, Faculty of Pharmaceutical Sciences, Government College University, Faisalabad, Faisalabad, Pakistan

[3] Department of Neurology, King Edward Medical University, Lahore, Pakistan

[4] Department of Endocrinology, King Edward Medical University, Lahore, Pakistan

[5] Department of Mathematics and Computer Science, Faculty of Science and Technology Prince of Songkla University, Pattani Campus, Thailand

[6] Department of Science and Humanities, National University of Computer and Emerging Science, Chiniot-Faisalabad Campus, Chiniot, Pakistan

[7] Department of Clinical Pharmacy, College of Pharmacy, King Khalid University, Abha, Saudi Arabia

Corresponding author
Shahid Shah,
shahid.waris555@gmail.com

## ABSTRACT

**Background and Objectives**. Postpartum depression (PPD) is prevalent among women after childbirth, but accessing mental healthcare for PPD is challenging. This study aimed to assess the treatment gap and barriers to mental healthcare access for women with PPD symptoms living in Punjab, Pakistan.

**Methods**. A multicenter cross-sectional study was conducted in five populous cities of Punjab from January to June 2023 by administering the questionnaire to the women using stratified random sampling. A total of 3,220 women in first 6 months postpartum were screened using the Edinburgh Postnatal Depression Scale. Of them, 1,503 women scored thirteen or above, indicating potential depressive disorder. Interviews were conducted to explore help-seeking behavior and barriers to accessing mental healthcare. Descriptive statistics along with nonparametric tests (*e.g.*, Kruskal–Wallis, Mann–Whitney U) were used and group differences were examined. Scatter plot matrices with fitted lines were used to explore associations between variables. Classification and regression tree methods were used to classify the importance and contribution of different variables for the intensity of PPD.

**Results**. Only 2% of women ($n = 33$) with high PPD symptoms sought mental healthcare, and merely 5% of women ($n = 75$) had been in contact with a health service since the onset of their symptoms. 92.80% of women with PPD symptoms did not seek any medical attention. The majority of women, 1,215 (81%), perceived the need for mental health treatment; however, 91.23% of them did not seek treatment from healthcare services. Women who recently gave birth to a female child had higher mean depression scores compared to those who gave birth to a male child. Age, education, and birth location of newborn were significantly associated ($p < 0.005$) with mean barrier scores, mean social support scores, mean depression scores and treatment gap. The

results of classification and regression decision tree model showed that instrumental barrier scores are the most important in predicting mean PPD scores.

**Conclusion**. Women with PPD symptoms encountered considerable treatment gap and barriers to access mental health care. Integration of mental health services into obstetric care as well as PPD screening in public and private hospitals of Punjab, Pakistan is critically needed to overcome the treatment gap and barriers.

# INTRODUCTION

Postpartum depression (PPD) refers to major depressive episode ''with peripartum onset'' if the onset of mood symptoms takes place either during pregnancy or in the four weeks following delivery (*Lu et al., 2023*). PPD is frequently confused with the brief ''baby blues'' that occur shortly after birth (*Parkash, 2021*). However, symptoms of PPD are more severe, stay longer, and may have the potential to cause postpartum psychosis (*Badri, 2022*). PPD is marked by low mood, loss of pleasure, decreased energy, reduced activity, marked functional impairment, decreased self-esteem, and thoughts of self-harm or suicide (*Lund & Town, 2016*).

The global prevalence of PPD is 17.2% (*Fish-Williamson & Hahn-Holbrook, 2023*). As with major depression, suicide associated with PPD has become the second leading cause of death among women (*Lindahl, Pearson & Colpe, 2005*; *Yu et al., 2021*). The suicide rates among women under 20 after giving birth or having an abortion are higher than in the same age group in the general population (*Gissler, Hemminki & Lonnqvist, 1996*).

The prevalence rate of PPD in Pakistan ranges from 28% to 63%, making it the highest among Asian countries (*Aliani & Khowaja, 2017*). A recent study showed a 67.96% prevalence of PPD in Pakistan (*Zulfiqar et al., 2023*), which is approximately four times higher than the worldwide prevalence rates. This can be attributed to several factors such as cultural norms (responsibility for household chores prioritized over own well-being, joint family system, postpartum customs), gender inequality, the stigma surrounding mental health, and socioeconomic factors such as poverty and limited access to healthcare services (*Gulamani, Shaikh & Chagani, 2013*; *Rahman, 2007*). Findings from other South Asian nations indicate concerning rates of suicide-related maternal deaths. For instance, studies from Sri Lanka and Nepal show that suicide accounts for 13% and 16% of maternal deaths, respectively (*Agampodi et al., 2014*; *Karki, 2011*). Moreover, children born to mothers with PPD in low and middle-income countries (LMICs) are more likely to have developmental issues (including cognitive issues, language issues, and academic delays), behavioral problems, and adverse impacts on health (*Agampodi et al., 2014*; *Azale, Fekadu & Hanlon, 2016*; *Surkan et al., 2011*).

Perceived social support is crucial in predicting and controlling self-related unpleasant life events and depression symptoms. Perceived social support is described as an individual's

subjective perception of whether or not their social network is supportive enough (*Nazari et al., 2020*). A recent study conducted in Pakistan revealed a significant association between social support and the development of PPD (*Riaz et al., 2023*). Exploratory studies have indicated that lower perceived social support during pregnancy increases the likelihood of developing depression symptoms during postpartum period (*Asselmann et al., 2020*; *Gan et al., 2019*; *Taylor et al., 2022*).

PPD is a clinical condition that frequently requires professional medical or/and psychological intervention (*Lackie et al., 2021*). There are a variety of management options available for PPD, with a strong evidence base to support the use of psychological approaches, including cognitive behavioral therapy (CBT) and interpersonal therapy, peer and partner support, and nondirective counseling (*Stewart & Vigod, 2019*). Despite numerous contacts with healthcare providers during the postpartum period, the majority of women with PPD do not seek treatment and are thus left undiagnosed (*Bina, 2014*). According to the literature, a variety of barriers influence help-seeking behaviors in women with PPD (*Cacciola & Psouni, 2020*; *Jones, 2019*; *Lackie et al., 2021*). Studies in high-income countries (HICs) showed that less than a quarter of women with PPD seek professional help (*Bina, 2014*; *McIntosh, 1993*). On the other hand, studies in similar cultures suggest that more than one-third of non-PPD individuals with major depressive disorder received treatment (*Moitra et al., 2022*). Therefore, the treatment gap for PPD appears to be larger than for non-PPD. Addressing treatment gap and barriers to mental healthcare access among women with PPD symptoms in Pakistan is critical for minimizing the negative impacts of PPD on maternal mental health and promoting healthier family environments. Our objective was to assess the treatment gap and barriers to mental healthcare access for women with PPD symptoms in Pakistan and to determine whether mean scores (depression, barriers, social support) vary significantly across sociodemographic and obstetric categories, to identify the strength and direction of relationship between depression scores, social support scores and barriers (scatter plot matrix) and development of a decision tree for classification of PPD scores among women with PPD symptoms.

## METHODOLOGY

### Study area

Punjab is the most populous province in Pakistan with a total population of about 127 million. In terms of land area, it ranks as the second-largest province. Lahore is the capital of Punjab and other major cities are Faisalabad, Rawalpindi, Multan, and Gujranwala. The healthcare structure of Punjab comprises primary, secondary, and tertiary/teaching care hospitals (THOS). Primary healthcare institutes consist of 2,500 Basic Health Units (BHUs) and 316 Rural Health Centers (RHCs). RHCs have 6,118 total working beds. Secondary healthcare institutes include 127 Tehsil Headquarter (THQ) hospitals with 7,170 functioning beds, and 26 District Headquarter (DHQ) hospitals have 7,416 total beds whereas 45 THOS are specialized care institutes and comprise 26,293 beds. In 2021, on average there were 321, 351, 118, 50, and 16 deliveries per month per THOS, DHQ, THQ, RHCs, and BHUs, respectively.

## Participants

The study included women aged 18–45, those within 6 months of postpartum, those who signed informed consent, and those with an EPDS score of more than or equal to ≥13. Women were excluded from the study if they experienced fetal demise or declined participation.

During the screening phase to identify women with PPD symptoms, data collectors approached a total of 3,340 individuals. While screening 3,340 individuals, 41 refused to further participate, of the rest of the 3,299 individuals, 79 out of 3,299 women didn't answer all items of the EPDS questionnaire. 3,220 women entirely completed the EPDS and 1,503 out of 3,220 had 13 or above scores. Those women ($n = 1,503$) who had positive PPD symptoms (13 or above scores, now met the inclusion criteria) were study participants and they were interviewed about their sociodemographic and obstetric characteristics, perceived social support, help-seeking behavior, their perceived need for care in the past six months, and barriers they face to access care.

The duration of the interview was between 18 to 20 min. Compensation was not provided to participants. This decision was made to maintain integrity by minimizing potential biases that could be introduced by offering compensation. Although the interviews proceeded smoothly, occasional interruptions occurred due to background noise and, in some cases, the presence of respondents' family members caused disturbances.

## Data collection instruments

In this study, socioeconomic and obstetric characteristics were considered basic demographic characteristics and independent variables, while social support, barriers to access to mental health care, and treatment gap were considered intervening variables.

The data collection instruments consisted of the following sections:

### The Edinburgh Postnatal Depression Scale

The Edinburgh Postnatal Depression Scale (EPDS) was used to screen for PPD symptoms. EPDS is a self-reported screening questionnaire that was designed for use in the postnatal period and serves as a valid screening tool for the detection of PPD (*Cox, Holden & Sagovsky, 1987*). This questionnaire comprises 10 questions, each scored on a 4-point Likert scale, with total scores ranging from 0 to 30. *Cox, Holden & Sagovsky (1987)* reported a split-half reliability of 0.88 and a standardized alpha coefficient of 0.87 for the EPDS and by using a cutoff score of 12 or 13, EPDS effectively identified cases of major PPD. Moreover, a score of 13 or above has greater sensitivity in South Asians and various other cultural settings (*Bhusal et al., 2016*; *Boyce, Stubbs & Todd, 1993*; *Cox, Holden & Sagovsky, 1987*; *Levis et al., 2020*; *Murray & Carothers, 1990*) and used to identify individuals with significant PPD symptoms (*Smith-Nielsen et al., 2021*). The Cronbach's alpha found in our study was 0.72 for EPDS.

### Socio-demographic and obstetrics characteristics

A structured proforma was utilized to gather socio-demographic and obstetrics-related information from the participants.

### *Multidimensional scale for perceived social support (MSPSS)*

Perceived social support from the participants' social networks, including spouses, family, and friends, was assessed using the MSPSS (*Zimet et al., 1988*). This instrument consists of 12 items, rated on a 7-point Likert scale. Higher scores indicate greater levels of perceived social support.

It has been proven to be psychometrically sound having good internal consistency (0.88) and test-retest stability (0.75 to 0.88) and strong factorial validity (*Zimet et al., 1988*). Subsequent studies, such as *Zimet et al. (1990)*, confirmed the instrument's factor structure and psychometric properties, including differential responses among participant groups, such as married *versus* never-married individuals and adolescents with varying levels of familial support (*Zimet et al., 1990*). The Urdu version of MSPSS has a good internal consistency as the Cronbach's alpha found in our study was 0.78 for MSPSS.

## General help-seeking questionnaire (GHSQ)

Help-seeking behavior was determined with the help of GHSQ (*Wilson et al., 2005*). The purpose of this scale is to measure the help-seeking intentions of individuals. The scale provides information regarding the help-seeking behavior of participants by formal and non-formal means. This is a nine-item 7-point Likert scale and categorizes help sources into two main categories: formal and informal (partners, friends, family, mental health professionals, physical health professionals, phone helpline, teachers, community, and no one) and problem type (personal-emotional and suicidal ideation) (*Wilson et al., 2005*). The GHSQ, when assessed in English, has demonstrated satisfactory psychometric characteristics, with internal consistency ranging from 0.67 to 0.90 and a test-retest reliability of 0.86.

To enhance the usefulness of the GHSQ, developers have suggested researchers to adapt the questionnaire by integrating specific types of sources of help relevant to their particular study (*Wilson et al., 2005*). In this study, the scale was adapted to assess actual help-seeking behavior rather than intended behavior, and the response categories were reduced to yes/no. Upon recommendations of researchers and to keep it aligned with the Pakistani cultural setting, this scale was adapted as follows: formal help-seeking sources were 1. General health professionals, 2. mental health specialists (*e.g.*, psychiatrists, psychologists), 3. others (traditional healers, and religious scholars/influencers), whereas non-formal sources for help-seeking involved parents, spouses, friends, and other relatives. Reliability of GHSQ was tested and a cronbach's alpha value of 0.72 was obtained.

### *Treatment gap*

The treatment gap was assessed by determining the disparity between the proportion of women needing mental health care and those utilizing mental healthcare services (*Kale, 2002*).

To determine the treatment gap, we assessed the gap between the perceived need to get care and actual treatment utilization (by GHSQ). Perceived need for treatment was determined using a dichotomous variable 'Perceived the need' from the Yes/No responses for each of the four categories of perceived need (information regarding illness and available

services, medication, counseling, skill training) (*Islam et al., 2022*; *Meadows et al., 2000*). Response options were "yes" or "no" against each type of four categories. To assess despite the need to access care, how many of them actually sought help, we used GHSQ. The GHSQ assessed how many women with PPD symptoms did or did not seek help from a general health professional or mental health professional in the past 6 months.

### Barriers to access to care evaluation

Barriers hindering access to mental healthcare services were assessed using the Barriers to Access to Care Evaluation (BACE) questionnaire (*Clement et al., 2012*). Participants were asked to rate the extent to which each item had prevented, delayed, or discouraged them from seeking or continuing treatment. Responses were recorded on a 4-point scale ranging from 0 (not at all) to 3 (a lot) (*Clement et al., 2012*). The mean scores were calculated for each subscale by averaging the total scores of all participants within each subscale. Cronbach's alpha value was 0.78 for BACE.

## Translation of questionnaires

For the translation of questionnaires, we followed the cross-cultural adaptation process (*Beaton et al., 2000*). We gathered questionnaires in English and translated them into Urdu.

First, a pharmacy post-graduate student and an individual with no medical background, both native Urdu speakers who are also fluent in English translated questionnaires into Urdu. Each of them produced a translation along with a report (mentioned uncertainties and an explanation of their choices) in written form. An independent researcher then joined and resolved any inconsistencies to produce a single forward-translated Urdu version. After this, a professional translator specializing in medical translation and a pharmacy post-graduate not previously involved in the initial translation translated this Urdu version back into English (one translation by each). Once again, an independent researcher compiled these back-translated versions and produced a pre-final version.

## Questionnaires validation
### Content validity

The investigators checked the survey instrument to ensure that the questions were clear, relevant, and easy to understand. The opinions and suggestions of researchers were considered such as when they suggested excluding "telephone helpline" from GHSQ as it was an uncommon source of seeking help in Pakistan. Similarly, in sociodemographic characteristics, "education" variable comprised of no formal education, middle, high school, bachelor, master or higher education, but researchers discussed and recommended including primary education to adequately capture the diversity of the study population (because girls in rural areas usually drop out after primary school due to a lack of high schools in villages). The questionnaire items were associated with the objectives of the study, demonstrating their content validity.

### Piloting the questionnaires

A pilot test was carried out with a specific sample to assess the appropriateness of the survey instrument. The respondents' feedback was not included in the final analysis.
### Reliability testing

We calculated Cronbach's alpha values for the estimation of the items' reliability. The cronbach's alpha values of instruments indicated good internal consistency.

## Procedure

A questionnaire-based multicenter cross-sectional study was carried out among women aged 18–45 years, during the first 6 months postpartum, who were willing to participate and provide informed consent, and had a PPD screening score of ≥13 on the EPDS. EPDS has a sensitivity of more than 80% in various cultural settings (*Atuhaire et al., 2023*; *Cox, Holden & Sagovsky, 1987*; *Husain et al., 2014*; *Shoaee et al., 2019*). This demonstrates that it is an acceptable way to correctly identify women having PPD. The study was carried out from January to June 2023. The screening and data collection process was conducted by a team consisting of two post-graduate pharmacy students and one post-graduate psychology student. Using stratified random sampling, the sample was chosen. First, we selected five major cities in Punjab, Pakistan: Faisalabad, Rawalpindi, Lahore, Multan, and Gujranwala. Next, we randomly selected seven public or private hospitals in each city, and finally, a total of 3,220 were screened for PPD symptoms from five cities.

We obtained the approval of the study from the Institutional Review Board, Government College University Faisalabad, Pakistan before initiating the study (Approval number: GCUF/ERC/196). The study met the guidelines outlined in the Helsinki Declaration. All patients provided informed verbal consent instead of written because the study did not involve any clinical intervention the patient's participation was clearly at minimum risk. Face-to-face administration of survey questionnaires was conducted in the native language, Urdu, using validated tools.

The Statistical Package for Social Sciences (SPSS, version 25.0) was used for data analysis. Descriptive statistical methods including frequency, percentages and Mean ± S.D were used to analyze the demographic and obstetric features of the study participants. The Mann–Whitney U and Kruskal–Wallis test were used to compare the mean score of treatment gap, social support and barriers across different demographic and obstetric characteristics. Scatter plot matrix with fitted linear regression line was used to present the relationships between treatment gap, stigma barriers, instrumental barriers, social support, and intensity of depression. Finally, the classification and regression tree (CRT) based decision tree method was used to classify the importance of different variables for the intensity of PPD.

## RESULTS

A total of 1,503 (46.67%) women scored 13 or above on the EPDS, indicating the presence of PPD symptoms. The mean EPDS score for the women was 18.99. A total of 3,220 women were screened for PPD symptoms; 46.67% (1,503) of the women with probably high PPD symptoms were study participants because they were assessed for sociodemographic and obstetric characteristics, barriers they faced to accessing care despite suffering from high PPD symptoms as well as to determine the treatment gap.

**Table 1 Socio- demographics characteristic of respondents ($n = 1,503$).**

| Variables | N (%) | Variables | N (%) |
|---|---|---|---|
| **Age** | | **Participant's employment status** | |
| 18 to 25 | 264 (18) | Employed | 150 (10) |
| 26 to 34 | 738 (49) | Not employed | 1,353 (90) |
| 35 to 44 | 483 (32) | **Husband's employment status** | |
| 45+ | 18 (1) | Employed | 1,062 (71) |
| **Area of residence** | | Not employed | 441 (29) |
| Rural | 165 (11) | **Gender of children** | |
| Urban | 1338 (89) | More male babies | 333 (22) |
| **Education** | | More female babies | 804 (53) |
| No formal education | 213 (14) | An equal number of male and female babies | 354 (24) |
| Primary | 336 (22) | No babies | 12 (1) |
| middle | 186 (12) | **Household monthly income** | |
| High school | 534 (36) | 25,000 PKR or less | 381 (25) |
| Bachelors | 198 (13) | 26,000–50,000 PKR | 906 (60) |
| Masters and above | 36 (02) | 50,000–1 lac PKR | 192 (13) |
| **Duration of marriage** | | More than 1 lac PKR | 24 (02) |
| less than 1 year | 30 (2) | **Living arrangements** | |
| 1 to 6 years | 576 (38) | Own house | 429 (29) |
| More than 6 years | 897 (60) | With the spouse's extended family | 1,074 (71) |

## Sociodemographic variables

The sociodemographic characteristics of the women in Table 1 demonstrated that approximately 49% of women with PPD symptoms were between the ages of 26 and 35. The majority of the women (89%) resided in urban areas. Education levels varied among the women, with 36% having attended high school, and 14% having no formal education. Employment status indicated that the majority of women were not employed (90%). Approximately 53% of women had more female babies. Regarding marriage duration, 60% of the women had been married for more than 6 years. Household income analysis revealed that approximately 60% of the women reported a monthly income between 26,000 PKR (approx. 90$) to 50,000 PKR (about 180$). In 2023, Pakistan had a GDP per capita of $1,568 (*Hussain, 2023*). Living arrangements indicated that 71% of the women were living with their spouses' extended family.

## Obstetric variables

The frequency and corresponding percentages of obstetric variables are presented in Table 2. Among the women, 77% had a gestational age of 9 months, indicating a full-term pregnancy. Approximately 61% of the newborns were female. The majority of the women (72%) gave birth in a government hospital. Only 17% of the women reported that the current pregnancy was their first. Additionally, 19% of the women had experienced a previous miscarriage.

**Table 2** Obstetric characteristics of respondents ($n = 1,503$).

| Variables | N (%) | Variables | N (%) |
|---|---|---|---|
| **Gestational age** | | **Type of delivery** | |
| 9 months | 1,164 (77) | Simple vaginal delivery | 318 (79) |
| Less than 9 months | 339 (23) | C - section | 1,185 (21) |
| **Gender of newborn** | | **Is it your first pregnancy** | |
| Female | 915 (61) | Yes | 252 (17) |
| Male | 588 (39) | No | 1,251 (83) |
| **Birth location of newborn** | | **Any previous miscarriage** | |
| Govt. Hospital | 1,086 (72) | Yes | 288 (19) |
| Private Hospital | 411 (27) | No | 1,215 (81) |
| Home | 6 (1) | | |

## Social support

The mean score for perceived social support was calculated by doing a sum across all 12 items on MSPSS and dividing by 12. Any mean score ranging from 3 to 5 is considered as moderate social support. The mean score for social support was calculated to be 4.376 which falls under the category of moderate social support. Table 3 represents the comparison of mean score of social support, depression, treatment gap, stigma barriers, attitudinal barriers, and instrumental barriers across all categories of sociodemographic and obstetric variables. However, the distribution of mean score for social support was found to vary significantly across different categories of age groups, educational status, gender of children, gender of the newborn, and first delivery status ($p < 0.05$), suggesting that there is a significant association between age groups, educational status, gender of children, gender of the newborn, and first delivery status and mean social support scores.

## Depression

The distribution of mean depression scores was not the same across various categories of sociodemographic and obstetric factors including education level ($p < 0.001$), the gender of children ($p < 0.001$), living arrangements ($p = 0.002$), gender of the newborn ($p = 0.03$), type of delivery ($p = 0.012$), and birth location of the newborn ($p = 0.004$) as shown in Table 3. These findings suggest the association of these sociodemographic and obstetric factors (education level, the gender of children, living arrangements, gender of the newborn, type of delivery, and birth location of the newborn) with depression scores.

## Treatment gap

The analysis revealed a significant treatment gap in accessing mental healthcare services among women with PPD symptoms. Table 4 elaborated that out of 1,503, only a small proportion of women sought medical attention for their PPD symptoms, with 33 women (2%) seeking specialized care from a mental health provider and 75 women (4%) were seeking help from a general physician. Even though a significant majority of the women 1,215 (81%) reported a perceived need for any type of mental health care, only 108 out of 1,215 were those who sought treatment either from general health professionals or mental health professionals. 91.23% of women with potentially high PPD symptoms who perceived
**Table 3** Comparison of score of treatment gap, social support and barriers across different demographic and obstetric characteristics ($n = 1,503$).

| Variables | Social support | | Depression | | Treatment gap | | Barriers | | | | | |
| --- | --- | --- | --- | --- | --- | --- | --- | --- | --- | --- | --- | --- |
| | | | | | | | Stigma barriers | | Attitudinal barriers | | Instrumental barriers | |
| | Mean (SD) | p-value | Mean (SD) | p-value | Mean (SD) | p-value | Mean (SD) | p-value | Mean (SD) | p-value | Mean (SD) | p-value |
| **Age** | | | | | | | | | | | | |
| 18–25 | 4.340 (0.9181) | | 18.57 (3.745) | | 1.34 (0.798) | | 13.33 (7.559) | | 13.50 (6.102) | | 7.15 (5.538) | |
| 26–34 | 4.342 (1.0157) | | 19.06 (3.986) | | 1.28 (0.920) | | 14.50 (7.736) | | 14.63 (6.683) | | 8.39 (5.366) | |
| 35–44 | 4.473 (0.9563) | 0.005 | 19.06 (3.455) | 0.071 | 1.26 (0.875) | 0.491 | 14.48 (7.578) | 0.047 | 14.35 (6.653) | 0.014 | 7.48 (5.390) | <0.001 |
| 45+ | 3.733 (0.7761) | | 19.67 (3.068) | | 1.50 (1.150) | | 11.00 (8.812) | | 11.83 (8.452) | | 4.17 (3.434) | |
| **Area of residence** | | | | | | | | | | | | |
| Rural | 4.364 (0.9819) | 0.488 | 19.33 (4.191) | 0.546 | 1.24 (0.833) | 0.560 | 15.78 (7.867) | 0.025 | 14.45 (6.999) | 0.894 | 9.02 (5.088) | 0.004 |
| Urban | 4.378 (0.9815) | | 18.94 (3.717) | | 1.30 (0.895) | | 14.06 (7.640) | | 14.29 (6.562) | | 7.68 (5.445) | |
| **Education** | | | | | | | | | | | | |
| No formal education | 4.203 (1.1207) | | 19.82 (3.67) | | 1.25 (0.853) | | 15.35 (8.147) | | 15.20 (6.964) | | 8.99 (5.601) | |
| Primary | 4.035 (1.0507) | | 19.01 (3.79) | | 1.18 (0.771) | | 17.38 (6.894) | | 16.24 (6.113) | | 10.62 (5.114) | |
| Middle | 4.450 (1.0158) | | 19.37 (3.58) | | 1.29 (1.009) | | 13.42 (7.433) | | 13.45 (6.389) | | 8.31 (4.999) | |
| High school | 4.585 (0.7333) | <0.001 | 18.26 (3.54) | <0.001 | 1.30 (0.879) | 0.001 | 12.66 (6.985) | <0.001 | 13.34 (6.196) | <0.001 | 5.87 (4.381) | <0.001 |
| Bachelors | 4.530 (1.0806) | | 19.61 (4.13) | | 1.55 (1.005) | | 13.21 (8.360) | | 13.68 (7.284) | | 6.88 (5.881) | |
| Masters and above | 4.275 (0.8872) | | 19.08 (4.52) | | 1.00 (0.586) | | 11.92 (9.545) | | 13.08 (7.85) | | 6.75 (7.291) | |
| **What is the gender of your children** | | | | | | | | | | | | |
| More male babies | 4.325 (1.0677) | | 18.58 (3.44) | | 1.35 (0.756) | | 13.95 (8.202) | | 14.70 (6.577) | | 7.58 (5.501) | |
| More female babies | 4.386 (0.9752) | | 19.17 (3.83) | | 1.29 (0.910) | | 14.64 (7.555) | | 14.65 (6.503) | | 8.35 (5.398) | |
| An equal number of male and female babies | 4.445 (0.8824) | <0.001 | 18.82 (3.86) | <0.001 | 1.23 (0.953) | 0.035 | 13.51 (7.507) | 0.079 | 13.00 (6.774) | <0.001 | 6.81 (5.299) | <0.001 |
| No babies | 3.125 (0.8400) | | 22.75 (3.16) | | 1.00 (0.793) | | 18.00 (1.954) | | 19.00 (2.663) | | 10.25 (3.166) | |

**Table 3** (*continued*)

| Variables | Social support | | Depression | | Treatment gap | | Barriers | | | | | |
|---|---|---|---|---|---|---|---|---|---|---|---|---|
| | | | | | | | Stigma barriers | | Attitudinal barriers | | Instrumental barriers | |
| | Mean (SD) | *p*-value | Mean (SD) | *p*-value | Mean (SD) | *p*-value | Mean (SD) | *p*-value | Mean (SD) | *p*-value | Mean (SD) | *p*-value |
| **Living arrangements** | | | | | | | | | | | | |
| Own house | 4.375 (1.0233) | 0.569 | 19.51 (3.98) | 0.002 | 1.47 (0.989) | <0.001 | 13.60 (8.615) | 0.077 | 14.03 (7.047) | 0.243 | 6.85 (5.827) | <0.001 |
| With Spouse's extended family | 4.377 (0.9644) | | 18.77 (3.66) | | 1.22 (0.835) | | 14.50 (7.263) | | 14.42 (6.426) | | 8.22 (5.203) | |
| **Gender of new born** | | | | | | | | | | | | |
| Female | 4.305 (1.0002) | 0.003 | 19.14 (3.82) | 0.030 | 1.19 (0.840) | <0.001 | 14.34 (7.694) | 0.384 | 14.22 (6.740) | 0.497 | 8.32 (5.559) | <0.001 |
| Male | 4.487 (0.9411) | | 18.73 (3.68) | | 1.44 (0.939) | | 14.10 (7.666) | | 14.44 (6.403) | | 7.06 (5.111) | |
| **Type of your delivery** | | | | | | | | | | | | |
| Simple vaginal delivery | 4.428 (0.9786) | 0.366 | 18.36 (3.15) | 0.012 | 1.38 (0.948) | 0.081 | 15.66 (6.636) | 0.001 | 15.12 (5.747) | 0.005 | 7.83 (4.491) | 0.535 |
| C-section | 4.363 (0.9819) | | 19.15 (3.90) | | 1.27 (0.871) | | 13.87 (7.898) | | 14.09 (6.808) | | 7.83 (5.647) | |
| **Birth location of newborn** | | | | | | | | | | | | |
| Home | 4.259 (1.0166) | | 19.09 (3.86) | | 1.24 (0.835) | | 15.77 (7.170) | | 15.34 (6.427) | | 8.88 (5.314) | |
| Government hospital | 4.663 (0.7946) | <0.001 | 18.63 (3.500) | 0.004 | 1.42 (1.003) | 0.027 | 10.16 (7.508) | <0.001 | 11.61 (6.364) | <0.001 | 5.10 (4.734) | <0.001 |
| Private hospital | 6.050 (0.2739) | | 23 (2.19) | | 1.00 (1.095) | | 18.50 (7.120) | | 11.50 (0.548) | | 5.00 (2.191) | |
| **Is it your first delivery** | | | | | | | | | | | | |
| Yes | 4.512 (0.9438) | 0.011 | 18.64 (3.51) | 0.169 | 1.55 (1.087) | <0.001 | 14.86 (7.896) | 0.289 | 14.33 (6.800) | 0.814 | 7.30 (5.190) | 0.117 |
| No | 4.349 (0.9867) | | 19.05 (3.82) | | 1.24 (0.834) | | 14.12 (7.635) | | 14.30 (6.573) | | 7.94 (5.463) | |

**Notes.**

*P* value for variables with two categories were calculated using Mann–Whitney U test and variable having more than two categories were tested using Kruskal–Wallis test.

the need to get treated could not seek any care from healthcare providers reflecting a high treatment gap. Moreover, of those who didn't perceive the need for any type of mental health care (21%), none of them used care from any healthcare professionals.

Table 3 demonstrates that the distribution of the treatment gap varied across various groups of education (*p = 0.001*), the gender of children (*p = 0.035*), living arrangements (*p < 0.001*), gender of the newborn (*p < 0.001*), type of delivery (*p = 0.012*), birth location of the newborn (*p = 0.027*) and first delivery status (*p < 0.001*).

## Barriers to access to mental healthcare

The distribution of stigma barriers, attitudinal barriers, and instrumental barriers across various sociodemographic and obstetric factors was assessed. The distribution of these barriers was not the same across all categories of age groups, educational status, and birth

**Table 4  Contribution of sources in seeking help by women with PPD symptoms ($n = 1503$).**

| Source of help | Yes N (%) | No N (%) |
|---|---|---|
| **Formal** | | |
| General health professional (any) | 75 (5) | 1,428 (95) |
| Mental health specialists (e.g., psychiatrists, psychologists) | 33 (2) | 1,470 (98) |
| Others (Traditional healers, religious leaders) | 72 (5) | 1,435 (95) |
| **Informal** | | |
| Husband | 726 (48) | 777 (52) |
| Friends | 186 (12) | 1,317 (88) |
| Parents | 300 (20) | 1,203 (80) |
| Other relatives | 213 (14) | 1,290 (86) |
| **Perceived need for mental health treatment** | **Yes N (%)** | **No N (%)** |
| Perceived the need for mental health treatment for PPD symptoms in the past 6 months. | 1,215 (81) | 288 (19) |

location of the newborn ($p < 0.05$). Variables, including age, education, and birth location of the newborn, were significantly associated with stigma, attitudinal and instrumental barriers scores.

Gender of children was significantly associated with mean attitudinal and instrumental barrier scores. Moreover, living arrangements were significantly associated with mean instrumental barrier scores.

Figure 1 depicts a decision tree for classifying PPD scores. Out of 1,503 women (node 0), 13% of women who had mean instrumental barrier scores >14.5 had the highest mean depression scores (22.354) whereas 87% of women who had mean instrumental barriers scores ≤14.5 had comparatively lower mean depression scores; however, 26.5% who had social support scores ≤4.25 (node 3), and instrumental barrier scores ≤14.5 had mean depression scores higher (19.401) than the other group (18.076) having mean social support scores > 4.25 (node 4).

Out of these 26%, 8% (node 6) of the women who had stigma barrier scores ≤14.5, social support scores ≤4.25; however, instrumental barrier scores >19.5 had mean depression scores of 20.575 which are higher than 18.66% of the women who had stigma barrier scores ≤19.5.

Importance of each variable (instrumental barriers, stigma barriers, attitudinal barriers, social support, treatment gap, gender of children, and age, in relation to depression) identified by CART analysis is presented in Fig. 2. Among these variables, instrumental barriers were considered the most important, based on their highest normalized importance value. The instrumental barrier scores are a highly crucial feature, suggesting that women with more barrier scores tend to have higher mean depression scores.

Figure 3 illustrates a scatter plot matrix to visualize the strength and direction of the relationship between variables. A negative relationship between social support and depression scores can be seen, as social support increases, mean depression scores decrease. There is also a negative relationship between social support and barrier (stigma,
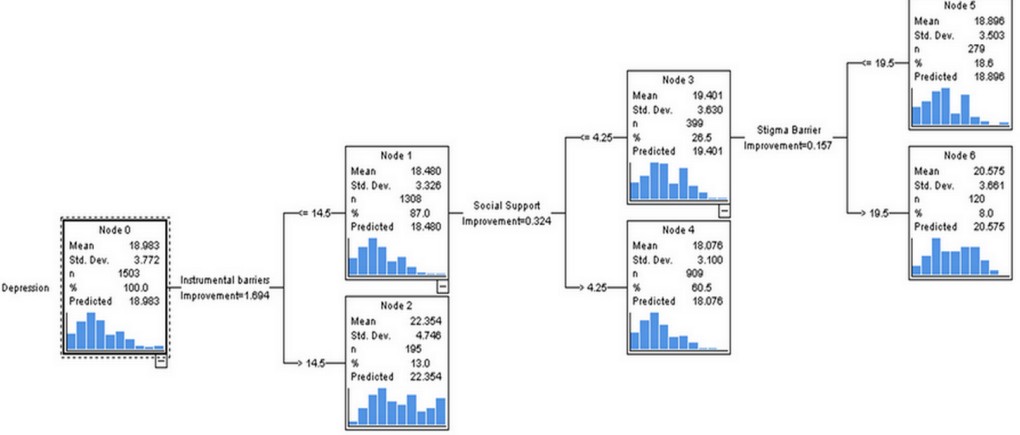

**Figure 1  Classification of intensity of PPD using decision tree with classification and regression tree (CRT) methods.**

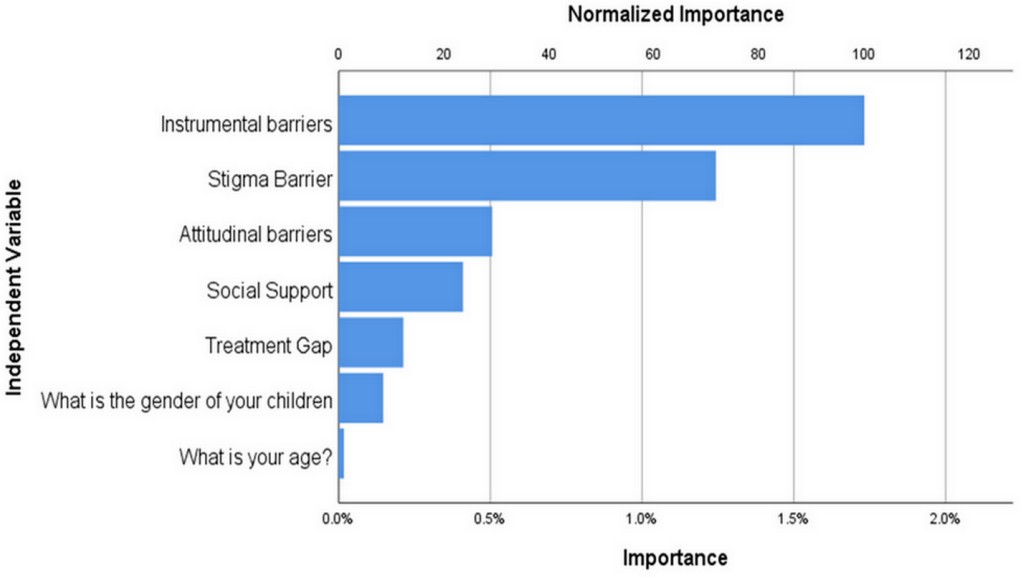

**Figure 2  Importance of variables in PPD using classification and regression (CRT) method.**

instrumental, attitudinal) scores, as mean social support scores increase, barrier scores decrease. A strong positive association between stigma and attitudinal barrier scores underscores the relationship between elevated stigma and higher levels of attitudinal barriers. Moreover, a moderate positive association between stigma and instrumental barrier scores can be visualized in the scatterplot matrix, which reinforces that increased stigma scores are linked to increased instrumental barrier scores. A weak positive relationship can be visualized between barriers (stigma, instrumental, attitudinal) and depression.

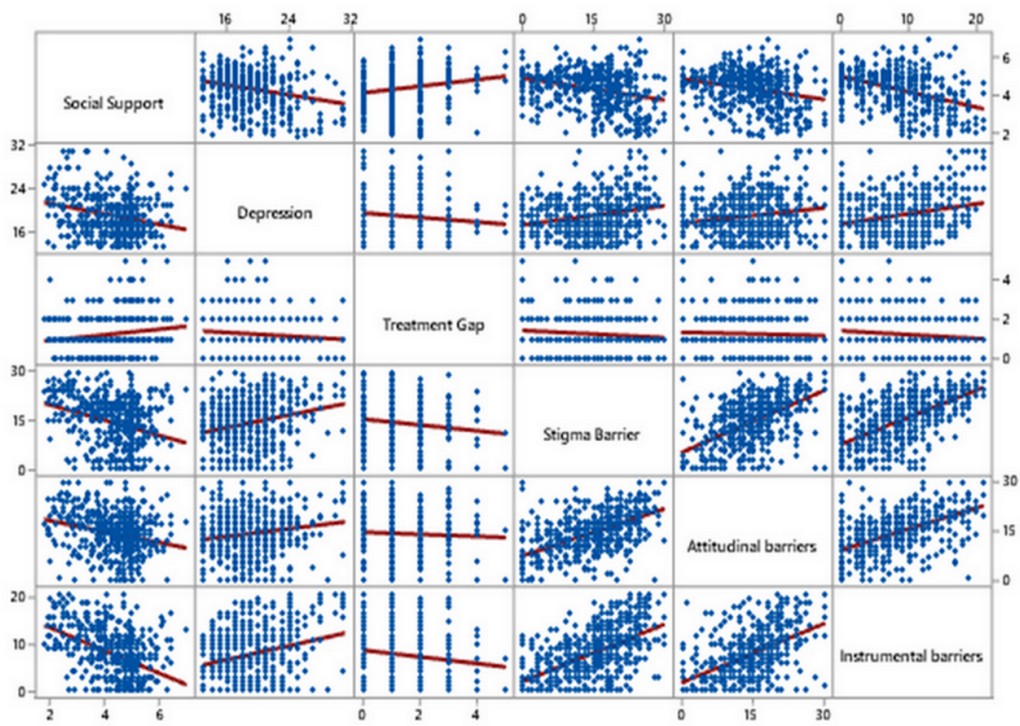

**Figure 3** Scatter plot matrix with fitted linear regression line for assessment of the relationship between treatment gap, social support and barriers.

## DISCUSSION

The present study aimed to investigate the treatment gap and barriers to accessing mental healthcare among women with PPD symptoms. Our findings revealed significant disparities in the utilization of mental healthcare services, with only a small proportion of women seeking help from mental health providers or general physicians. This indicates a substantial treatment gap, highlighting the need for interventions to improve access and address the barriers faced by these women. A similar study conducted in Ethiopia reported a high treatment gap among PPD women, with about 87% of women having PPD did not seek treatment from any healthcare service. The compatibility of our findings with the Ethiopian study is evident, as both investigations identified a substantial treatment gap among women affected by PPD (*Azale, Fekadu & Hanlon, 2016*). The observed treatment gap in both Pakistan and Ethiopia might be attributed to the insufficient prioritization of mental health within the overall healthcare infrastructure and systems. Additionally, in our study the high proportion of women expressing a need to seek help from health services for their PPD symptoms suggests a favorable level of acceptability for integrating mental health care into maternal health care platforms within public and private hospitals in Punjab, Pakistan. Among informal sources of help, approximately half of the women preferred to seek help from their husbands, and one-fifth of the women sought help from their parents.

EPDS scores varied across different categories of sociodemographic and obstetric factors. Education level, gender of children, living arrangements, gender of the newborn, type of delivery, and birth location of the newborn were significant factors associated with mean depression scores. Studies conducted in the United States, Sweden, India, and Ethiopia revealed a significant association between the risk of PPD and sociodemographic and obstetric factors such as education level, mode of birth, and number of children (*Kim & Dee, 2018*; *Mollard et al., 2016*; *Shelke & Chakole, 2022*; *Silverman et al., 2017*; *Zeleke et al., 2021*). The presence of similar findings in LMICs as well as HICs could be attributed to the universal influence of certain sociodemographic and obstetric factors on maternal mental health during the postpartum period. Social support was identified as a crucial factor. The mean social support score indicated moderate support, and its distribution varied significantly across different categories of age, educational status, gender of children, and gender of the newborn, birth location of the newborn, and first delivery status. The scatter plot matrix illustrates a negative relationship between social support and depression scores can be seen, as social support increases, mean depression scores decrease. Findings of the study were consistent with other studies where increased levels of support were associated with lower PPD scores (*Chien, Tai & Yeh, 2012*; *Edwards et al., 2012*; *Morikawa et al., 2015*; *Pao et al., 2019*). This consistent pattern across diverse cultures emphasizes the strength of the relationship between social support and depression, highlighting the universal importance of strong social networks in promoting mental well-being. Stigma, attitudinal, and instrumental barriers were identified, with age, education, and birth location of the newborn impacting these barriers. Women aged 26 to 34 showed higher stigma, attitudinal, and instrumental barrier scores than women in other age groups (18–24, 35–44, and 45+). This finding suggests that women in this age group may encounter specific challenges or attitudes that impede their willingness to seek help for PPD symptoms. Educational status also played a significant role, with women who had no formal education experiencing higher attitudinal and instrumental barriers than women who had high school, bachelor's, or master's degrees. This finding highlights the importance of education in promoting mental health literacy and empowering women to recognize and seek appropriate support for PPD symptoms. Moreover, women residing in rural areas faced higher instrumental barriers compared to their urban counterparts. A study conducted in Maharashtra reported that the majority of women participants mentioned insufficient time and lack of financial resources as barriers to seek metal healthcare. This finding underscores a broader trend observed not only in the context of PPD but also in relation to other mental health challenges, indicating that women in rural areas often encounter significant instrumental obstacles when seeking mental healthcare. Notably, instrumental barriers emerged as the most important factor contributing to depression symptoms. Our results underscore the necessity of addressing logistical challenges and improving access to mental healthcare services. A study in Israel revealed barriers at different levels that collectively hindered women's access to treatment for PPD. Such as lack of PPD knowledge, insufficient social support, the absence of healthcare providers, economic constraints, and residing in rural areas where there is a lack of basic facilities. These barriers were found at the individual level, the family level, organizational level, community level, and public policy level, respectively

(*Alfayumi-Zeadna et al., 2019*). Therefore, interventions should focus on reducing barriers, increasing social support, and raising awareness to effectively address PPD.

To overcome the treatment gap and barriers to access care, screening of women before and after childbirth has proven effective (*Flynn et al., 2006*; *MacArthur et al., 2002*). Moreover, it has been reported that women who had depression screening during the peripartum phase in community or hospital settings were almost four times more likely to seek treatment (*Flynn et al., 2006*). Therefore, the effectiveness of screening women before and after childbirth in hospitals in Pakistan can significantly increase treatment-seeking behaviors among women experiencing peripartum depression. World Health Organization (WHO) has suggested integrating mental health care into maternal care services for the mental well-being of mothers (*World Health Organization (WHO), 2022*); however, lack of mental health professionals in Pakistan and a lower-than-internationally recommended GDP allocated for mental health pose challenges (*Siddiqui, 2021*; *Zhu et al., 2014*). To tackle the shortage of mental health professionals, the Mental Health Gap Action Programme (mhGAP), developed by the WHO has been employed by various South Asian countries to train non-specialist healthcare providers (*Keynejad, Spagnolo & Thornicroft, 2021*). This aims to enhance access to perinatal mental healthcare across different environments by integrating mental health services into primary care and there is a critical need to implement mhGAP in Pakistan. Providing integrated mental healthcare into obstetric care and screening during peripartum and postpartum phases at THOS, DHQs, THQs, RHCs and private hospitals will probably improve their access to mental healthcare and reduce the barriers.

## LIMITATIONS

This multicenter cross-sectional design of the study limits us from establishing causal relationships between variables. Additionally, the study was conducted in specific geographical regions, limiting the generalizability of the findings to other populations with different socio-cultural contexts. Moreover, the study focused solely on the perspectives of women with PPD, excluding the viewpoints of healthcare providers and other stakeholders involved in mental healthcare.

## CONCLUSION

The present study highlighted the substantial treatment gap and barriers faced by women with PPD symptoms in accessing mental healthcare services. Our findings revealed that sociodemographic and obstetric factors, including age, education, living arrangements, and birth-related variables, significantly contributed to these barriers. Addressing the challenges of PPD requires targeted interventions and policy changes. Integrating mental health services into obstetric wards of public (RHCs, DHQs, THQs and THOS) and private hospitals, raising awareness through policy-driven seminars, and empowering women to seek timely help can enhance access to mental healthcare and reduce stigma.

### Funding

The Deanship of Scientific Research at King Khalid University funded this work through large group Research Project under grant number RGP2/486/44. The funders had no role in study design, data collection and analysis, decision to publish, or preparation of the manuscript.

### Grant Disclosures

The following grant information was disclosed by the authors:
Deanship of Scientific Research at King Khalid University: RGP2/486/44.

### Competing Interests

The authors declare there are no competing interests.

### Author Contributions

- Aqsa Sajjad conceived and designed the experiments, performed the experiments, authored or reviewed drafts of the article, and approved the final draft.
- Shahid Shah conceived and designed the experiments, prepared figures and/or tables, authored or reviewed drafts of the article, and approved the final draft.
- Ghulam Abbas performed the experiments, authored or reviewed drafts of the article, and approved the final draft.
- Ayesha Aslam performed the experiments, prepared figures and/or tables, authored or reviewed drafts of the article, and approved the final draft.
- Fawad Randhawa analyzed the data, authored or reviewed drafts of the article, and approved the final draft.
- Haris Khurram analyzed the data, authored or reviewed drafts of the article, and approved the final draft.
- Abdullah Assiri analyzed the data, authored or reviewed drafts of the article, and approved the final draft.

### Human Ethics

The following information was supplied relating to ethical approvals (i.e., approving body and any reference numbers):

The approval of the study from the Institutional Review Board, Government College University Faisalabad, Pakistan has been approved before initiating the study (Approval number: GCUF/ERC/196)

### Ethics

The following information was supplied relating to ethical approvals (i.e., approving body and any reference numbers):

This research was approved by the IRB GC university Faisalabad (GCUF/ERC/196).

### Data Availability

The raw data is available in the Supplemental File

## Supplemental Information

Supplemental information for this article can be found online at http://dx.doi.org/10.7717/peerj.17711#supplemental-information.

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
