# Peer review of "Treatment gap and barriers to access mental healthcare among women with postpartum depression symptoms in Pakistan"

_PeerJ, doi:10.7717/peerj.17711_

## Round 0.1 · original submission · Major Revisions

The reviewers sent detailed comments and annotated PDF's so that if you methodically and thoroughly address all the comments of both reviewers the manuscript will be much improved and will most probably be accepted.

I found your study important and would like to see it revised.

Let me just repeat some of the very important points made by the reviewers:

1, You need to inform the reader about PPD in Pakistan and put it in context of PPD in the world.

2, You need to tell more about your sample and about your methodology. As written it is puzzling. Both reviewers made constructive suggestions as to how you may rewrite your methodology section - use all the comments of both reviewers to do so/

Good Luck!
Ada

Reviewer 1 ·

Basic reporting

I enjoyed reading this article that relates to a very important and relevant issue in the Pakistani context, and worldwide.
Attached is my review of the article. As you can see from my comments, I recommend publication after major revisions.

The manuscript “Treatment Gap and Barriers to Access Mental Healthcare Among Women with PPD Symptoms”, examines the difficulties women in Pakistan, from both urban and rural areas, encounter when seeking help for postpartum depression (PPD) symptoms. This research contributes important insights into this significant public health issue.
In a questionnaire-based multicentre cross-sectional study design, the authors gathered socio-demographic and obstetric data from 1503 women aged 18-45, those within six months of postpartum, with an EPDS score of ≤13. Those women were then interviewed about their attempts to seek help and the barriers they faced when trying to access mental healthcare.
Using suitable statistical methods, the authors identified a substantial gap in the utilization of mental healthcare services. Only a small fraction of women sought help from healthcare professionals. This finding is pertinent not only to Pakistan but also to other low-to-middle-income countries globally. It provides evidence that urges policymakers to dedicate more resources to mental health services.
However, the paper requires some improvements before it can be considered for publication. The following suggestions are provided in the order they appear in the paper, not by their significance:
1. The title - I suggest informing the readers in the title that this study was conducted in Pakistan.
2. Introduction:
A. The introduction presents a detailed and up-to-date literature review. To make it easier for the reader, the authors should: 1) Divide the introduction into subtopics (for example PPD, resilience factors such as perceived social support, treatment gap); 2) relate, in separate sections, to what is known from research findings concerning PPD around the world (for instance, Yu et al. 2021), and those found in Pakistan; 3) indicating the country in which the cited study was conducted, so the reader will easily understand that this finding has been repeated from studies in different countries or cultures.
B. The authors should explicitly state the research hypotheses.
3. Methodology:
C. There is room to add a subsection with a brief description of the 'Study Areas'.
D. There is a need to refer to the following issues: 1) translation of the questionnaires from English to local national languish; 2) duration of the interview 3) compensation for the subjects, and 4) special problems that arose during the interviews.
E. Please add data concerning the reliability and validity of the questionnaires used in the research.
F. Validity of the questionnaire - Please, add examples to the claim that " The researchers’ views and comments were taken into consideration" (L. 146-147).
4. Results
G. which percentage of the 1503 women who participated in the study represents the total population who participated in the study. (L. 164).
H. Add values in dollars to the monetary amounts displayed in local currency (L. 174) and include information regarding GDP per capita.
5. Discussion
I. A more extensive summary of the main findings is required.
J. "Additionally, in our study the high proportion of women…" (L. 247). A comparison with similar data in Ethiopia is requested.
K. Kindly compare the findings not only with those from urban California (L.254-256) but also with those from another low-income country. This comparison will further substantiate the claim (L. 256-259).”
L. Chein et al 2012 etc. (L. 263-264). These studies were conducted in diverse cultural contexts, a point that deserves emphasis.
M. There is a need to include a subchapter that explores the implications of the research findings and proposes evidence-based, effective strategies to face the treatment gap and accessibility to mental health services among women who suffer from PPD (see for instance Qi W et al. (2022). Effects of family relationship and social support on the mental health of Chinese postpartum women. BMC pregnancy and childbirth, 22(1), 65.‏).
N. It is necessary to expand and clarify how the research findings on barriers to mental health care in Israel are related to the sentence: "…barriers collectively hinder the access of women to PPD treatment" (L. 279).

Experimental design

The research fulfils the aims and scope of the Journal. The Research questions, which are relevant and meaningful, are elaborated. The research has been conducted in compliance with the prevailing ethical standards in the field. The methods used are well described and allow replication of the study.

Validity of the findings

All underlying data have been provided. The findings are validated and have a significant contribution to the field by adding cross-cultural evidence about the treatment gap and barriers to mental health care for women suffering from PPD among the participants in the survey.

The conclusions are appropriately stated, connected to the original question investigated, and limited to those supported by the results.

Annotated reviews are not available for download in order to protect the identity of reviewers who chose to remain anonymous.

·

Basic reporting

This is a study on an important topic with important findings, and as such deserves to be published. However, there are currently significant problems in how the study is presented and explained in my opinion, and these need to be addressed.
The English is good overall but needs some polishing for clarity.
Background is sufficient, including references.
Structure needs attention, specifically in the Methods section. Please include three subsections: Participants, Instruments and Procedure (including statistical analyses).
An overarching aim is presented ("to address the treatment gap and barriers to mental healthcare access for women with PPD symptoms"). Please provide broader and more specific aims, and if possible hypotheses.

Experimental design

The study's aims and scope are suitable to the journal.
The research question should be better defined and it should be better explained how the study fills a gap in our knowledge. The Pakistani context should be stressed more than it currently is.
My major criticism is methodological. The participants section states that there were 3340 respondents, of whom 41 refused to participate (so how were they respondents???) and 79 did not complete all the questionnaires. However, inclusion criteria include a score of 13 or above on the EPDS and the Results seem to present the outcomes of analyses run only with this sub-sample of respondents.
This is first of all very confusing, since numbers remain somewhat unclear throughout. Analyzing the data only of these women also seems to waste good data from a valuable comparison group. I therefore recommend:
- stating the n clearly in the text, tables and figures throughout.
- Including the women who scored below 13 on the EPDS as a comparison group where possible, to add validity to what we learn about the women with probably PPD - otherwise at least some findings could reflect characteristics of all postpartum women in Pakistan.
- I am not convinced that Figures 1 and 2 are necessary and would advise presenting a full correlation table instead, unless there are clear statistical advantages to the way results have been presented.
- Please explain in the Methods section how the "importance" of the variables in Figure 3 were determined.
I have included other comments in the PDF file attached.

Validity of the findings

The findings seem valid for the women scoring 13 or above on the EPDS only.
Conclusions are well stated but should be more specifically linked to the hypotheses and analyses.

Additional comments

Please see annotated PDF file.

---

## Round 0.2 · Minor Revisions

Your revised manuscript has now been reviewed by one of the original reviewers, who has made several suggestions for improvement and recommended minor revisions.

I join the reviewer and ask that you carefully go over the annotated manuscript and include all the editing that they have suggested; Some of it is linguistic, but some of the comments are substantive - for instance the comment that you did not validate the questionnaires. A lot of the comments are in the abstract and methods section - please go over them methodically they will greatly improve your paper.

In addition, there is an absence of comparison to other low-income countries for the rates of PDD and for the treatment gap, In your first rebuttal letter you explained that there was no suitable research in other such countries, but this is patently incorrect.

Here are some references you may wish to use in the comparison that is lacking in the manuscript at present:

Okunola, T. O., Awoleke, J. O., Olofinbiyi, B., Rosiji, B., Olubiyi, A. O., & Omoya, S. (2022). Predictors of postpartum depression among an obstetric population in South-Western Nigeria. Journal of reproductive and infant psychology, 40(4), 420-432.
Shelke, A., & Chakole, S. (2022). A review on risk factors of postpartum depression in India and its management. Cureus, 14(9).
Zeleke, T. A., Getinet, W., Tadesse Tessema, Z., & Gebeyehu, K. (2021). Prevalence and associated factors of post-partum depression in Ethiopia. A systematic review and meta-analysis. PloS one, 16(2), e0247005.

Please take care to implement all these suggestions in your next revision

Ada

·

Basic reporting

The English is acceptable - I have made many editing suggestions in the annotated file. I have also made suggestions for the improvement of the basic structure.
Perhaps I missed something, but I could not find the revised figures and tables.

Experimental design

The design is now clearer. It remains unclear how stigma, attitudinal and instrumental barriers were calculated. This should be presented in the Methods section.

Validity of the findings

Findings seem valid.

Additional comments

Overall the manuscript has been improved and most comments addressed.
I have attached the manuscript with my suggestions for further editing and revision.

---

## Round 0.3 · accepted · Accept

Thank you for submitting this manuscript to PeerJ and working through the revisions. It is a very interesting study. The situation of women giving birth in Pakistan is distressing as seen through this study.

Reviewer 1 ·

Basic reporting

The comments I made in an earlier phase has been addressed adequately.
This is an excellent manuscript that deserves to be published as is.

Experimental design

The research questions are well defined, meaningful, and important.

Validity of the findings

All underlying data have been provided; they are robust, statistically sound, & controlled.

The conclusions are clearly articulated, directly related to the original research question, and confined to supporting results.